# Increasing Egg Consumption at Breakfast Is Associated with Increased Usual Nutrient Intakes: A Modeling Analysis Using NHANES and the USDA Child and Adult Care Food Program School Breakfast Guidelines

**DOI:** 10.3390/nu13041379

**Published:** 2021-04-20

**Authors:** Yanni Papanikolaou, Victor L. Fulgoni

**Affiliations:** 1Nutritional Strategies, Nutrition Research & Regulatory Affairs, 59 Marriott Place, Paris, ON N3L 0A3, Canada; 2Nutrition Impact, Nutrition Research, 9725 D Drive North, Battle Creek, MI 49014, USA; vic3rd@aol.com

**Keywords:** US National Health and Nutrition Examination Survey (NHANES), eggs, children, breakfast, CACFP, usual intakes, nutrient adequacy

## Abstract

The objective of the current modeling analysis was three-fold: (1) to examine usual nutrient intakes in children when eggs are added into dietary patterns that typically do not contain eggs; (2) to examine usual nutrient intakes with the addition of eggs in the Child and Adult Care Food Program (CACFP) school breakfast; and (3) to examine nutrient adequacy when eggs are included in routine breakfast patterns and with the addition of eggs to the CACFP school breakfast program. Dietary recall data from the National Health and Nutrition Examination Survey 2011–2016 (children aged 1–18 years-old; *n* = 9254; CACFP *n* = 159) were used in the analysis. The usual intakes of pantothenic acid, riboflavin, selenium, and vitamin D increased ≥10 percent (relative to the baseline values) with the addition of one egg at breakfast. The usual intakes of protein and vitamin A at breakfast were also increased by more than 10 percent compared to the baseline values with the addition of two eggs. Similar outcomes were observed with the addition of eggs to the CACFP school breakfast. The percent of children above the adequate intake for total choline increased to 43.6 and 57.8% with one and two eggs, respectively, compared to 22.6% at the baseline. The addition of eggs at breakfast can contribute to nutrient intakes and overall dietary adequacy and play a role in public health initiatives aimed at increasing the intake of under-consumed nutrients and nutrients of concern.

## 1. Introduction

The importance of breakfast has been well documented in the scientific literature; however, large portions of Americans are still not eating breakfast [1]. Consumer survey data show that as many as 31 million Americans do not consume any foods before the lunch period. In children, the highest incidence for breakfast skipping is linked to adolescents, with 14% of 13- to 17-year-olds reporting no food consumption before lunch [2]. Indeed, this raises public health concerns, as breakfast consumption in children has previously been shown to be a key contributor to nutrient density and overall diet quality. Data from the UK National Diet and Nutrition Survey Rolling Programme demonstrated that children who frequently consume breakfast had significantly greater intakes of dietary fiber, folate, calcium iron, and iodine relative to those who skipped breakfast. Further analyses showed that higher percentages of breakfast consuming children met nutrient-specific recommendations compared to children who skipped breakfast [3]. Data from the US National Health and Nutrition Examination Survey (NHANES) has identified associations with breakfast skipping and all-cause mortality, in that skipping breakfast was associated with an 87% increased risk for cardiovascular mortality and an 19% increased risk for all-cause mortality [4]. A recent meta-analysis of eight observational studies also showed that breakfast skipping was related to an elevated risk of type 2 diabetes [5].

In the US, public health and nutrition initiatives have focused on providing access to breakfast to vulnerable populations, with the creation of the Child and Adult Care Food Program (CACFP)—a nutrition-focused meal program that is operated at the US federal level, wherein reimbursements are provided to eligible children and adults enrolled for care at approved centers, including family day care homes, group child care centers, emergency shelters, and afterschool programs [6]. Detailed criteria have been established for what can be served at meals, and they can be food specific. For example, yogurt must contain no more than 23 g of sugar per each 6 ounce serving, or 3.83 g of sugar per ounce. Cooking methods are also specified, such that healthy cooking includes roasting, baking, sautéing, and steaming, but does not allow for deep fat frying [6]. The USDA has updated and revised the CACFP meal patterns to ensure that children and adolescents have access to healthy, balanced meals at breakfast and throughout the day [6]. Effective in 2017, requirements in the CACFP meal patterns allowed for meat and meat alternatives to be served in place of the entire grains component at breakfast [7]. Meat and meat alternatives at breakfast include beans, cheese, eggs, lean meat/poultry/fish, nut butters, tofu, and yogurt, with standards set in place for minimum serving amounts at breakfast that are dependent on age group. For example, the minimum amount of eggs that can be served in place of the entire grains component at breakfast for children 1- to 5- and 6- to 18-years-old are one-fourth and one-half of a large egg at breakfast, respectively, to a maximum of three times/week. Similarly, the 2020–2025 Dietary Guidelines for Americans (DGA 2020–2025) also promote a variety of protein foods (which include eggs), particularly when consumed with limited sodium, solid fat, and added sugar [8]. The Dietary Guidelines Scientific Advisory Committee (DGAC 2020) includes eggs as an introductory food for infants and as part of a healthy dietary pattern in all children [9]. As eggs have been previously established to be a nutrient-dense food, with one 50 g serving (i.e., one egg) contributing several bioactive components and essential nutrients that are routinely underconsumed with current eating patterns.

Recent published data in children show the dietary benefits of eggs within overall eating patterns [10,11,12]; however, limited data are available that have examined the outcomes associated with the addition of eggs to the breakfast meal. Therefore, the purpose of the current modeling analysis was three-fold: (1) to examine usual nutrient intakes in children when eggs are added into dietary patterns that typically do not contain eggs; (2) to examine usual nutrient intakes with the addition of eggs in CACFP school breakfast; and (3) to examine nutrient adequacy when eggs are included in routine breakfast patterns.

## 2. Experimental Section

Data for the current analyses were obtained from NHANES, a US cross-sectional, nationally representative sample, which includes children ages 1- to 18-years-old. NHANES is a continuous study governed by the Centers for Disease Control and Prevention (CDC) and samples free-living Americans (i.e., noninstitutionalized) every 2 years [13]. The requisite ethical protocols, including informed consent from the study participants, were previously obtained, approved, and documented by the CDC ethic boards. Three datasets were combined for the present study (NHANES 2011–2012, 2013–2014, and 2015–2016) [14,15,16]. Data for the nutrients examined are from the U.S. Department of Agriculture (USDA) Food and Nutrient Database for Dietary Studies (FNDDS) database for NHANES [17]. The FNDDS databases determine food and beverage nutrient values in What We Eat in America (WWEIA), which represents the dietary intake component of NHANES. The collection procedure for WWEIA involves use of the Automated Multiple Pass Method (AMPM), representing a dietary collection tool that provides a valid, evidence-based approach for gathering data for national dietary surveys. The accuracy, effectiveness, and efficiency of the AMPM method have been comprehensively described and previously published [18].

### 2.1. Nutrient Intake Modeling Scenarios

NHANES 2011–2016 dietary recall and questionnaire data were obtained for subjects aged 1-to 18-years-old, with exclusions for incomplete data (*n* = 9254 subjects). Data interpreted to be reliable was comprised of 24 h of completed recalled dietary data. Pregnant and lactating females were not included in the analysis. Nutrient intakes were examined for the following modeling scenarios:

Baseline: No changes to the typical breakfast pattern

Model 1: Addition of one egg to breakfast when no eggs were typically consumed at breakfast

Model 2: Addition of two eggs at breakfast when no eggs were typically consumed at breakfast

Model 3: Addition of one egg at breakfast when breakfast is CACFP compliant

Model 4: Addition of two eggs to breakfast when breakfast is CACFP compliant

Breakfast was determined as CACFP compliant if fluid milk, vegetables/fruit, and grains quantities satisfied the following USDA criteria:

Milk: 4 oz unflavored whole milk (age 1 year), 4 oz unflavored 1% or skim milk (age 2 years), 6 oz unflavored 1% or skim milk (age 3–5 years), 8 oz unflavored 1% or skim milk or flavored skim milk (age 6–18 years)

Fruit/Vegetables: 0.25 oz eq (age 1–2 years), 0.5 oz eq (age 3–18 years)

Grain: 0.5 oz eq (age 1–5 years), 1.0 oz eq (age 6–18 years)

The definition of egg consumption within the modeling scenarios included 1 oz eq eggs from whole eggs, boiled eggs, or poached eggs. Nutrient profiles for 1 oz eq egg were used to determine nutrient intakes when eggs were added to breakfast in the four modeling scenarios. Egg intake was determined by FNDDS food codes defined in the appropriate WWEIA category (with the exclusion of ‘egg substitutes’, and ‘other poultry eggs’). The present study did not include egg-containing or grain-based mixed dishes (i.e., egg-containing sandwiches, breakfast burritos, and all egg-containing bakery foods, including cakes, breads, cookies, and biscuits).

### 2.2. Methodolgy

Statistical procedures were completed with the employment of SAS software (Version 9.4, SAS Institute, Cary, NC, USA). The investigation used day 1 dietary survey weights to develop nationally representative estimates for children and adolescents, along with adjustment for the complex sample design of the database. Usual intake was estimated using the National Cancer Institute method, version 2.1 and has been published previously by our group [12,19]. Means (±standard errors) for daily nutrient intakes were determined for baseline scenarios and all modeling scenarios. For select nutrients where an established estimated average requirement (EAR) is available, nutrient inadequacy was calculated (i.e., percent below EAR) for all children. Similarly, where nutrients have an authoritative adequate intake (AI), nutrient adequacy was determined (i.e., percent above AI) for all children where a sufficient sample size was available. Nutrient inadequacies/adequacies’ percentages, standard errors, and lower/upper and 95th confidence levels for select nutrients were determined. When 95th confidence levels did not overlap, we deemed the changes to be meaningful.

## 3. Results

### 3.1. Nutrient Usual Intakes with the Addition of Eggs at Breakfast

Current daily nutrient usual intakes are presented in Table 1. Additionally, the usual intakes for macro- and micro-nutrients when modeling the various scenarios are presented in Table 1. In modeling the addition of one and two eggs at breakfast, the usual intakes of total choline, lutein + zeaxanthin, docosahexanoic acid, vitamin D, vitamin B5, and cholesterol increased by greater than 10 percent (relative to the baseline). Furthermore, the addition of one and two eggs at breakfast increased the usual intakes of cholesterol, lutein + zeaxanthin, docosahexaenoic acid, and total choline by ≥20 percent over baseline (see Table 1).

### 3.2. Nutrient Usual Intakes with the Addition of Eggs to CACFP School Breakfast

Adding one and two eggs to the CACFP compliant breakfast resulted in meaningful changes to usual nutrient intakes, relative to baseline nutrient intake. The addition of one and two eggs to the CACFP school breakfast resulted in a ≥20 percent increase in usual intake of cholesterol, docosahexaenoic acid, and total choline, while a ≥20% over the usual intake was observed with lutein + zeaxanthin. Adding one egg to the CACFP breakfast resulted in a ≥10% increase in vitamin A, vitamin D, and pantothenic acid (Table 2).

### 3.3. Nutrient Inadequacy and Adequacy with the Addition of Eggs at Breakfast

When compared to the baseline intakes, the addition of eggs at breakfast resulted in meaningful outcomes in nutrient inadequacy and nutrient adequacy outcomes. The percent above the adequate intake for total choline increased from 43.6 and 57.8% with one and two eggs, respectively, compared to 22.6% at baseline. Similar benefits were observed for vitamin A and vitamin D, such that the percentage below the estimated average requirement decreased with the addition of one and two eggs (Table 3).

### 3.4. Percent of Children above Dietary Guidelines’ Recommended Cutoffs for Daily Saturated Fat and Sodium

When considering the various dietary models, the differences were less than 10 percent in comparison with the baseline values for the percent of children above the cutoff of 10% of caloriesfrom saturated fat in the daily diet. Similarly, differences were below 10 percent (relative to the baseline) for the percentage above the sodium cutoff of 2300 mg/day, except when modeling the addition of two eggs at breakfast, wherein the percentage of children above the sodium cutoff increased by more than 10% versus baseline.

## 4. Discussion

As breakfast habits in childhood are predictors for eating behaviors in adulthood [20], establishing healthy breakfast standards in the early years becomes a critical public health initiative. The current study modeled usual nutrient intakes after various scenarios where eggs were added to the routine breakfast pattern and the CACFP compliant school breakfast. The results showed that adding one or two eggs at breakfast in American children increased several nutrient intakes by greater than 10 to 20 percent in comparison to baseline. Specifically, modeling the addition of one and two eggs at breakfast, the usual intakes of total choline, lutein + zeaxanthin, and docosahexanoic acid increased by greater than 20 percent compared to baseline usual intakes. The current analysis also provides evidence that the addition of eggs at breakfast can play a role in nutrient adequacy and/or inadequacy. Indeed, the percent above the adequate intake for total choline increased from 43.6 and 57.8% with one and two eggs, respectively, compared to 22.6% at the baseline. Further, adding eggs to breakfast moderately decreased the percentage of children below the estimated average requirement.

The recent release of the 2020 Dietary Guidelines Advisory Committee (DGAC 2020) scientific report included food and nutrition recommendations for birth to 24-month-olds. Eggs have been recommended as part of the first complimentary foods introduced to infants at 4- to 6-months of age (representing a historic first in authoritative food and nutrition recommendations), and eggs have been recommended as part of a healthy dietary pattern in older children [9]. The DGAC 2020 also highlighted public health challenges in that several nutrients are under-consumed by all Americans (≥1 years-old), including magnesium, choline, and vitamins A, C, E, and K. However, vitamin D, calcium, dietary fiber, and potassium have been identified as nutrients of public health concern, implying that the underconsumption of these nutrients has been associated with adverse health outcomes. The DGAC 2020 further stated that children aged 9- to 14-years-old have a “constellation of potential nutritional risk factors that are considered a public health challenge”, with girls and boys demonstrating inferior intakes of choline, magnesium, and phosphorus [9]. The current modeling analysis provides evidence to support the addition of eggs within current breakfast patterns. Indeed, when considering underconsumed nutrients in Americans, the addition of one and two eggs at breakfast led to increases greater than 10% (relative to the baseline) in the usual intakes for total choline, vitamin D, and vitamin A—three nutrients identified by the 2020 DGAC as underconsumed nutrients in all Americans [9]. Furthermore, the current nutrient adequacy analysis showed that the modeling of one and two eggs at breakfast increased the percentage of children above the AI for total choline. The DGAC 2020 classified choline as an underconsumed nutrient in particular, as choline has been extensively documented as a key nutrient in neurodevelopment, metabolism and physiological functions [21]. Eggs are a leading food source for choline in the diet, such that one serving (i.e., 50 g, large egg) provides 147 mg of dietary choline [22]. NHANES data supports the conclusion that the majority of Americans, including children and adolescents, are not meeting the established recommendations for choline intake, thus creating a potential public health concern regarding choline intake [23]. A recent review highlighted the limited data availability on the usual intake of choline in different age groups and populations, highlighting research on children to address gaps in the scientific literature [24].

The current analysis has shown that the addition of one and two eggs at breakfast leads to a greater than usual intake of lutein + zeaxanthin. While a dietary reference intake has not been established for lutein or zeaxanthin, authoritative guidance recommends consumption of vegetables (dark green vegetables in particular), as they represent a rich source of lutein and zeaxanthin. However, dietary guidance has reported low mean intakes of vegetables across all age groups relative to the recommendations, and children from 1 to 18 years of age all fall below dark green vegetable recommendations in their consumption habits [8,9]. Eggs are a dietary source for bioavailable lutein and zeaxanthin, with a 50 g egg contributing approximately 250 µg lutein + zeaxanathin [22]. Lutein and zeaxanthin are carotenoids, with accumulating data supporting a beneficial role in eye health and risk reductions in terms of the prevalence of ocular diseases [25,26,27].

As has been documented previously in observational studies similar to the present analysis, our modeling study has limitations characteristic of epidemiological research. These limitations have previously been discussed and outlined in numerous publications [28,29,30]. Nonetheless, the strengths and robust nature of the NHANES database provides a valuable insight into American dietary patterns and associated health outcomes.

## 5. Conclusions

To our knowledge, the current data represent the first study in children that examined the modeling outcomes following the addition of eggs to breakfast. The addition of one or two eggs at breakfast in 1- to 18-year-old children increased several positive nutrients, including total choline, lutein + zeaxanthin, docosahexanoic acid, vitamin D, vitamin A, and pantothenic acid, by greater than 10 percent in comparison to the usual baseline intakes. Moreover, the usual intakes of total choline, lutein + zeaxanthin, and docosahexaenoic acid were elevated by greater than 10 percent relative to the baseline with the addition of two eggs at breakfast. The current study also showed that adding eggs to breakfast patterns can play a role in nutrient adequacy where meaningful increases were seen in the percentage of children above the established adequate intake for total choline relative to the baseline. The addition of eggs to breakfast can contribute to nutrient intakes and overall dietary adequacy and play a role in public health initiatives aimed at increasing the intake of under-consumed nutrients and nutrients of concern.

## Figures and Tables

**Table 1 nutrients-13-01379-t001:** Effects of the inclusion of eggs in breakfast on the usual intake of nutrients among children aged 1–18 years-old; gender combined; US National Health and Nutrition Examination Survey (NHANES) 2011–2016; *n* = 9254.

	Usual Intake					
Nutrients	Baseline	SE	Breakfast + 1 Egg	SE	Breakfast + 2 Eggs	SE
	(Scenario 0)		(Scenario 1)		(Scenario 2)	
Calcium (mg)	1023	10	1039	11	1054	11
Carbohydrates (g)	244	2	244	2	245	2
Cholesterol (mg)	214	2	319 **	3	421 **	3
Dietary Fiber (g)	13.9	0.1	13.9	0.1	13.8	0.1
Energy (kcal)	1851	15	1891	15	1931	14
Energy from Sat Fat (%)	11.7	0.1	11.9	0.1	12.1	0.1
Folate DFE (mcg)	499	6	510	6	519	6
Iron (mg)	13.7	0.2	14.2	0.2	14.7	0.2
Lutein + Zeaxanthin (mcg)	769	17	907 **	17	1060 **	19
Magnesium (mg)	231	2	234	2	238	2
Niacin (mg)	20.8	0.2	20.8	0.2	20.9	0.2
Pantothenic Acid (mg)	28.3	0.3	31.5 *	0.3	34.8 **	0.4
PFA 18:2 (Octadecadienoic) (g)	13.7	0.2	14.1	0.2	14.6	0.2
PFA 18:3 (Octadecatrienoic) (g)	1.3	0.02	1.4	0.02	1.4	0.02
PFA 20:5 (Eicosapentaenoic) (g)	0.01	0.0003	0.01	0.0003	0.01	0.0003
PFA 22:6 (Docosahexaenoic) (g)	0.02	0.0008	0.04 **	0.001	0.06 **	0.001
Phosphorus (mg)	1247	10	1302	10	1359	10
Potassium (mg)	2146	17	2185	16	2222	16
Protein (g)	66.8	0.6	70.2	0.6	73.7*	0.6
Riboflavin (mg)	1.9	0.02	2.1 *	0.02	2.2 *	0.02
Selenium (mg)	93.9	1	102 *	1	111 **	1
Sodium (mg)	2924	27	3046	27	3165	27
Thiamin (mg)	1.5	0.01	1.5	0.02	1.53	0.01
Total Choline (mg)	248	2	314 **	3	379 **	3
Total Fat (g)	69.8	0.6	72.3	0.7	75	0.7
Total Monounsaturated Fat (g)	23.6	0.3	24.6	0.3	25.6	0.3
Total Polyunsaturaed Fat (g)	15.4	0.2	15.9	0.2	16.4	0.2
Total Saturated Fat (g)	24.6	0.3	25.4	0.3	26.3	0.3
Total Sugars (g)	115	1	115	1	115	1
Trans Fat (g)	1.9	0.03	1.9	0.03	2	0.03
Vitamin A RAE (mcg)	594	8	640	9	683 *	9
Vitamin B12 (mcg)	4.7	0.1	4.9	0.1	5.1	0.1
Vitamin B6 (mg)	1.7	0.02	1.7	0.02	1.8	0.02
Vitamin C (mg)	74.2	1.7	74.2	1.7	74.1	1.7
Vitamin D (D2+D3) (mcg)	5.7	0.1	6.3 *	0.1	6.8 *	0.1
Vitamin E, as alpha-tocopherol (mg)	6.9	0.1	7.2	0.1	7.5	0.1
Vitamin K (mg)	64.8	1.3	64.7	1.3	64.9	1.3
Zinc (mg)	9.8	0.1	10.1	0.1	10.5	0.1

* ≥10% higher vs. baseline; ** ≥20% higher vs. baseline; SE = standard error; PFA = polyunsaturated fatty acids.

**Table 2 nutrients-13-01379-t002:** Effect of inclusion of eggs in the Child and Adult Care Food Program (CACFP) school breakfast on the usual intake of nutrients among children aged 1–18 years-old; gender combined; NHANES 2011–2016; *n* = 159.

	Usual Intake					
Nutrients	Baseline CACFP	SE	CACFP + 1 Egg	SE	CACFP + 2 Eggs	SE
	(Scenario 3)		(Scenario 4)		(Scenario 5)	
Calcium (mg)	1326	53	1341	53	1356	52
Carbohydrates (g)	252	12	252	11	253	11
Cholesterol (mg)	234	20	339 **	17	444 **	14
Dietary Fiber (g)	16.3	1	16.3	1	16.3	1
Energy (kcal)	1879	76	1924	75	1969	73
Energy from Sat Fat (%)	11.1	0.5	11.5	0.5	11.7	0.5
Folate DFE (mcg)	555	37	565	37	576	36
Iron (mg)	15.1	1	15.7	1.1	16.2	1.1
Lutein + Zeaxanthin (mcg)	995	126	1122 *	113	1279 **	121
Magnesium (mg)	274	12	277	12	280	12
Niacin (mg)	20.7	1.2	20.8	1.2	20.8	1.2
Pantothenic Acid (mg)	36.2	1.7	39.5	1.7	42.8 *	1.7
PFA 18:2 (Octadecadienoic) (g)	13.1	0.7	13.5	0.7	14.1	0.7
PFA 18:3 (Octadecatrienoic) (g)	1.3	0.1	1.3	0.1	1.3	0.1
PFA 20:5 (Eicosapentaenoic) (g)	0.01	0.0009	0.01	0.0009	0.01	0.0009
PFA 22:6 (Docosahexaenoic) (g)	0.02	0.005	0.04 **	0.003	0.05 **	0.003
Phosphorus (mg)	1523	59	1579	61	1632	59
Potassium (mg)	2683	112	2721	116	2758	110
Protein (g)	76.1	3.3	79.8	3.3	83.1	3.2
Riboflavin (mg)	2.4	0.1	2.5	0.1	2.6	0.1
Selenium (mg)	108	6	116	6	125	6
Sodium (mg)	2862	128	2990	125	3104	121
Thiamin (mg)	1.5	0.1	1.6	0.1	1.6	0.1
Total Choline (mg)	312	16	378 **	14	446 **	10
Total Fat (g)	66.3	3.4	68.9	3.3	71.4	3.3
Total Monounsaturated Fat (g)	21.9	1.3	22.9	1.3	23.9	1.3
Total Polyunsaturaed Fat (g)	14.7	0.8	15.3	0.8	15.9	0.8
Total Saturated Fat (g)	23.6	1.4	24.5	1.4	25.3	1.3
Total Sugars (g)	122	6	121	6	121	6
Trans Fat (g)	1.8	0.1	1.8	0.1	1.8	0.1
Vitamin A RAE (mcg)	766	39	805	35	862 *	36
Vitamin B12 (mcg)	5.7	0.2	5.9	0.2	6.1	0.2
Vitamin B6 (mg)	1.7	0.1	1.8	0.1	1.8	0.1
Vitamin C (mg)	104	13	104	13	104	13
Vitamin D (D2+D3) (mcg)	8.7	0.5	9.2	0.5	9.8 *	0.5
Vitamin E, as alpha-tocopherol (mg)	7.2	0.6	7.4	0.6	7.7	0.6
Vitamin K (mg)	79.3	13.8	79.4	13.8	79.4	13.7
Zinc (mg)	11	0.5	11.3	0.5	11.7	0.5

* ≥10% higher vs. baseline; ** ≥20% higher vs. baseline; SE = standard error; PFA = polyunsaturated fatty acids.

**Table 3 nutrients-13-01379-t003:** Effects of inclusion of eggs in breakfast on nutrient inadequacy (% below the estimated average requirement) or nutrient adequacy (% above the adequate intake) among children 1–18-years-old; gender combined; NHANES 2011–2016; *n* = 9254.

	% of Children Above Adequate Intake or Below Estimated Average Requirement											
Nutrients	Baseline	SE	LCL	UCL	Breakfast + 1 Egg	SE	LCL	UCL	Breakfast + 2 Eggs	SE	LCL	UCL
	(Scenario 0)				(Scenario 1)				(Scenario 2)			

**% Above Adequate Intake**												
PFA 18:3 (Octadecatrienoic) (g)	75.6	1.5	72.7	78.6	76.7	1.5	73.8	79.6	77.6	1.5	74.7	80.6
Total Choline (mg)	22.6	0.9	20.9	24.3	43.6 *	1.2	41.4	45.9	57.8 **	2.8	52.4	63.2
**% Below Estimated Average Requirement**												
Phosphorus (mg)	13.6	1.1	11.4	15.8	10.8 *	1.1	8.7	12.9	8.3 **	0.9	6.5	10.1
Protein (g)	0.5	0.2	0.1	0.8	0.3	0.1	0.003	0.5	0.2	0.1	0.0	0.3
Riboflavin (mg)	0.9	0.3	0.3	1.4	0.5	0.2	0.1	0.9	0.3	0.1	0.01	0.6
Vitamin A RAE (mcg)	23.2	1.4	20.6	25.9	18.7 *	1.4	16.1	21.5	14.8 **	1.2	12.5	17.1
Vitamin D (D2+D3) (mcg)	92	0.7	90.6	93.4	89.6 *	0.9	88.0	91.3	86.1 **	0.9	84.2	87.9

* significantly different vs. baseline (Scenario 0); ** significantly different vs. Scenario 1; SE = standard error; PFA = polyunsaturated fatty acids; LCL = 95% lower confidence level; UCL = 95% upper confidence level.

## Data Availability

NHANES was used to conduct the present analysis and is a publicaly available dataset. The NHANES data can be found at https://wwwn.cdc.gov/nchs/nhanes (accessed on 20 April 2021).

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
