# Peer review of "Increasing Egg Consumption at Breakfast Is Associated with Increased Usual Nutrient Intakes: A Modeling Analysis Using NHANES and the USDA Child and Adult Care Food Program School Breakfast Guidelines"

_nutrients, 2021, doi:10.3390/nu13041379_

Round 1

Reviewer 1 Report

The paper outlines the modelling outcomes of adding one or two eggs to the breakfast of 1 to 18 year old children.  There was a greater than 10 percent increase in total choline, lutein + zeaxanthin, docosahexanoic acid, vitamin D, vitamin A and pantothenic acid intake compared with their usual intake. The study showed that adding eggs to these children’s usual breakfast patterns can contribute to nutrient intakes and overall dietary adequacy and play a role in public health initiatives aimed at increasing intake of under-consumed nutrients and nutrients of concern.

  • Only one sentence in the discussion appeared to be incomplete as outlined in bold on line 200 below....

The DGAC 2020 also highlighted public health challenges in 198

that several nutrients are under-consumed by all Americans (≥1 years-old), including 199

magnesium, choline, vitamin A, C, E and K. However, vitamin D, calcium, dietary fiber 200

and potassium. DGAC 2020 further stated that children aged 9- to 14-years-old have a 201

“constellation of potential nutritional risk factors that are considered a public health chal-202

lenge”, with girls and boys demonstrating inferior intakes of choline, magnesium, and 203

phosphorus [9]. The current modeling analysis provides evidence to support the addition 204

Author Response

Dear Reviewer,

Thank you for taking the time to review our paper and provide feedback. We have provided answers and comments to your questions and suggestions below. Please let us know if you have any further questions or suggestions. The authors’ answers are in bold font.

Sincerely,

Yanni Papanikolaou and Victor L. Fulgoni

Reviewer’s Question/Comment: Only one sentence in the discussion appeared to be incomplete as outlined in bold on line 200 below....

The DGAC 2020 also highlighted public health challenges in that several nutrients are under-consumed by all Americans (≥1 years-old), including magnesium, choline, vitamin A, C, E and K. However, vitamin D, calcium, dietary fiber 200 and potassium... DGAC 2020 further stated that children aged 9- to 14-years-old have a “constellation of potential nutritional risk factors that are considered a public health challenge”, with girls and boys demonstrating inferior intakes of choline, magnesium, and phosphorus [9].

Authors’ Response: Thank you for identifying this error in the manuscript. We have revised the section to read as follows (see Lines 201-202):

The DGAC 2020 also highlighted public health challenges in that several nutrients are under-consumed by all Americans (≥1 years-old), including magnesium, choline, vitamin A, C, E and K. However, vitamin D, calcium, dietary fiber and potassium have been identified as nutrients of public health concern, implying that underconsumption of these nutrients has been associated with adverse health outcomes. DGAC 2020 further stated that children aged 9- to 14-years-old have a “constellation of potential nutritional risk factors that are considered a public health challenge”, with girls and boys demonstrating inferior intakes of choline, magnesium, and phosphorus [9].

Please reach back with any further questions and/or comments. Thank you.

Reviewer 2 Report

Page 2, line 86: Three datasets are combined to form a single dataset of 9,254 individuals for this work. Are the individuals unique for each of the three studies? Or is it possible that the same individual has contributed data in more than one of the three NHANES dataset? If so, do the authors take that into account in their data analysis?

Page 2, lines 103-107: “when no eggs typically consumed at breakfast”: do all 9,254 individuals of the NHANES 2011-2016 dataset satisfy this? If not, the exact number should be mentioned in the manuscript. Similarly, the number of CACFP compliant breakfasts should also be mentioned.

Table 1: the interpretation of the results is not very clear, particularly to someone not fairly experienced with similar studies and the NHANES datasets. For example, first row and first column: Calcium baseline is 1023 mg; is this an average across the NHANES 2011-2016 individuals? Or only over the ones with “no eggs typically consumed at breakfast”? Similarly, what does a standard error of 10 mg mean in this context? (this comment applies to table 2 as well).

Table 3: Do the percentages of this table refer to the 9,254 individuals or a subset of them? Additionally, while the percentage of individuals above the threshold conveys a lot of information, statistical significance should also be tested.

Finally, it is my understanding that this work suggests the addition of 1 or 2 eggs in breakfast as a means to increase nutrient intake. However, the rationale about why focusing on this particular item (i.e. egg) is not very clear in the manuscript. It would also be interesting to see a comparison between some alternative products.

Author Response

Dear Reviewer,

Thank you for taking the time to review our paper and provide feedback. We have provided answers and comments to your questions and suggestions below. Please let us know if you have any further questions or suggestions. The authors’ answers are in bold font.

Sincerely,

Yanni Papanikolaou and Victor L. Fulgoni

Reviewer’s Question/Comment: Page 2, line 86: Three datasets are combined to form a single dataset of 9,254 individuals for this work. Are the individuals unique for each of the three studies? Or is it possible that the same individual has contributed data in more than one of the three NHANES dataset? If so, do the authors take that into account in their data analysis?

Authors’ Response: Thank you for this important question regarding NHANES. We are confident that individuals from one NHANES cycle to another are unique. It is highly unlikely that an individual in one NHANES study would be in different NHANES study as the sampling is done in geographic blocks, with NHANES not repeating the same geographic blocks. In the unlikely event that an individual from one NHANES study be found in a different NHANES study, we as researchers would never be able to identify that individual, given the use of SEQN—SEQN refers to the sequence number and is a unique identifier for each observation/participant in NHANES.

Reviewer’s Question/Comment: Page 2, lines 103-107: “when no eggs typically consumed at breakfast”: do all 9,254 individuals of the NHANES 2011-2016 dataset satisfy this? If not, the exact number should be mentioned in the manuscript. Similarly, the number of CACFP compliant breakfasts should also be mentioned.

Authors’ Response: You are correct. ‘When no eggs are typically consumed at breakfast”, 9,254 participants were included in the analysis.

Reviewer’s Question/Comment: Table 1: the interpretation of the results is not very clear, particularly to someone not fairly experienced with similar studies and the NHANES datasets. For example, first row and first column: Calcium baseline is 1023 mg; is this an average across the NHANES 2011-2016 individuals? Or only over the ones with “no eggs typically consumed at breakfast”? Similarly, what does a standard error of 10 mg mean in this context? (this comment applies to table 2 as well).

Authors’ Response: For Tables 1 and 2, the data presented are usual intakes for each nutrient (i.e., usual intakes were determined using the National Cancer Institute’s method described below). The usual intakes means were determined using the collective data from NHANES 2011-2016. Further, the standard error provides the reader with information on how accurate the mean of the sample is—since the SE for calcium is small (i.e., 10 mg), it is likely that the usual mean intake is accurate.

Reviewer’s Question/Comment: Table 3: Do the percentages of this table refer to the 9,254 individuals or a subset of them? Additionally, while the percentage of individuals above the threshold conveys a lot of information, statistical significance should also be tested.

Authors’ Response: The percentages presented in Table 3 refer to the sample size of 9,254 individuals. We have revised the table to include 95% lower and upper confidence levels to provide a measure of significance. We have also added additional text to the methodology to address your comments (see Lines 134-137). Based on your feedback, we believe that these changes have improved the manuscript. Thank you.

Reviewer’s Question/Comment: Finally, it is my understanding that this work suggests the addition of 1 or 2 eggs in breakfast as a means to increase nutrient intake. However, the rationale about why focusing on this particular item (i.e. egg) is not very clear in the manuscript. It would also be interesting to see a comparison between some alternative products.

Authors’ Response: Your understanding is correct in that this study was conducted to assess how addition of eggs is associated with nutrient intakes. The rationale for conducting this modeling analysis was to add to the existing body of literature, particularly since eggs represent a nutrient dense food option for children and adolescents. Further, prior to initiating this work, we noticed that there was a lack of usual nutrient intake data linked to egg consumption. Thus, we used the National Cancer Institutes methodology to estimate usual intake. The following overview is provided by the National Cancer Institutes website (Usual Dietary Intakes | EGRP/DCCPS/NCI/NIH (cancer.gov):

Usual dietary intake is the long-term average daily intake of a nutrient or food. The concept of usual intake is important because dietary recommendations are intended to be met over time and diet-health hypotheses are based on dietary intakes over the long term. Consequently, it is the usual intake that is often of most interest to policy makers -- when they want to know the proportion of the population at or below a certain level of intake -- or to researchers -- when they want to examine relationships between diet and health. However, until recently, sophisticated efforts to capture this concept have been limited at best.”

Within our manuscript, we have listed the following rationale for conducting this study (see Lines 72-78):

Recent published data in children show the dietary benefits of eggs within overall eating patterns [10,11,12], however, limited data are available that have examined the outcomes associated with the addition of eggs at the breakfast meal. Therefore, the purpose of the current modeling analysis was three-fold: 1) to examine usual nutrient intakes in children when eggs are added into dietary patterns that typically do not consume eggs; 2) to examine usual nutrient intakes with the addition of eggs in CACFP school breakfast; and 3) to examine nutrient adequacy when eggs are included in routine breakfast patterns.

Furthermore, we are in progress of a follow-up study that would expand on the current analyses to include alternative products and comparisons to eggs.

Please let us know if you have further comments and/or questions.

Round 2

Reviewer 2 Report

I would like to thank the authors for their responses; the methodology is now much clearer. I have no further comments, except for some minor ones regarding type-setting (e.g. page 2 seems to be aligned to right).